# Native-Valve *Aspergillus* Endocarditis: Case Report and Literature Review

**DOI:** 10.3390/antibiotics12071190

**Published:** 2023-07-14

**Authors:** Claudio Caroselli, Lorenzo Roberto Suardi, Laura Besola, Alessandro Fiocco, Andrea Colli, Marco Falcone

**Affiliations:** 1Infectious Diseases Unit, Department of Clinical and Experimental Medicine, Azienda Ospedaliera Universitaria Pisana, University of Pisa, 56126 Pisa, Italy; caroselli.claud@gmail.com (C.C.); marco.falcone@unipi.it (M.F.); 2Cardiac Surgery Unit, Department of Surgical, Medical and Molecular Pathology and Critical Care, Azienda Ospedaliero Universitaria Pisana, University of Pisa, 56126 Pisa, Italy; laura.besola@ao-pisa.toscana.it (L.B.); alessandro.fiocco@ao-pisa.toscana.it (A.F.); andrea.colli@unipi.it (A.C.)

**Keywords:** endocarditis, fungal, *Aspergillus*, negative blood culture endocarditis, difficult-to-treat infection

## Abstract

*Aspergillus* endocarditis represents the second etiological cause of prosthetic endocarditis following *Candida* spp. On the other hand, native-valve endocarditis due to *Aspergillus* are anecdotally reported with increasing numbers in the last decade due to new diagnostic technologies such as *polymerase chain reaction* (PCR) on samples like valve tissue or entire blood. We performed a review of the literature presenting one case report observed at Pisa University Hospital. Seventy-four case reports have been included in a period between 1950–2022. Immunocompromised status (patients with solid tumor/oncohematological cancer or transplanted patients) was confirmed to be the main risk factor for this rare opportunistic infection with a high rate of metastatic infection (above all, central nervous system) and mortality. Diagnosis relies on serum galactomannan and culture with PCR on valve tissue or whole blood. Cardiac surgery was revealed to be a life-saving priority as well as appropriate antifungal therapy including b-liposomal amphotericin or new triazoles (isavuconazole). The endocarditis team, facing negative blood culture endocarditis affecting an immunocompromised patient, should investigate this difficult-to-treat pathogen.

## 1. Introduction

*Aspergillus* is a ubiquitous mold that has the ability to cause broad spectrums of clinical manifestations both in healthy and immunocompromised humans [1,2].

Endocarditis caused by *Aspergillus* spp. (AE) is an uncommon disease with a high mortality rate that usually develops in patients with a predisposing condition, such as immunocompromised status or previous cardio-thoracic surgery. The diagnosis and treatment of the infection is highly challenging, while the disease often foreshadows a poor prognosis [3].

In this article, we describe a case of native-valve *Aspergillus* endocarditis (NVAE) occurring in a middle-aged Caucasian woman without previous cardiac medical history nor known immunocompromised status and we performed a review of the available literature.

## 2. Case Report Description

In November 2021, a 74-year-old Caucasian woman was transferred from a secondary general hospital to Pisa University Hospital presenting with acute heart failure and cardiogenic shock in order to be evaluated by the cardiac surgery team. At first evaluation, a huge aortic valve vegetation was discovered upon cardiac ultrasonographic findings.

With regards to her medical history, she was a former smoker with untreated arterial hypertension and dyslipidemia, with no history of intravenous drug use.

Eight months earlier, she had undergone an atypical endoscopic resection of the lower lobe of the left lung for an adrenocorticotropic hormone (ACTH)-secreting neuroendocrine neoplasia that caused Cushing’s syndrome with secondary diabetes.

The patient arrived at Pisa University Hospital sedated and in endotracheal intubation; laboratory data showed normal white blood cells count (8470/mcl) with absolute neutrophilia and lymphopenia (7340/mcl and 640/mcl, respectively) and increased plasma levels of C-reactive protein (CRP, 11.63 mg/dL) and procalcitonine (PCT, 5.53 ng/mL) as well as increased levels of nt-proBNP (17,531 ng/L), high sensitivity troponins (1084 ng/L), CPK-MB (22.2 U/L) and myoglobin (238 g/L). Microbiological sampling was performed and two sets of blood cultures were collected. Subsequently, an empirical antibiotic therapy with meropenem (3 g per day) and vancomycin (2 g per day) was started and the patient immediately (<3 h since transferal) underwent aortic valve replacement with a bioprosthesis Perimount Magna Ease 21 mm (Edward Lifesciences Corporation, Irvine, CA, USA); the native aortic valve was sent to the laboratory to perform microbiological culture.

On the 5th post-operative day, the patient was extubated with good clinical response. Inflammatory markers (CRP and PCT) showed a positive reduction; therefore, antimicrobial therapy with meropenem and vancomycin was continued. The patient was still treated in the ICU.

On the 7th post-operative day, the microbiology lab informed that a strain of *Aspergillus fumigatus* grew from a flap of the native aortic valve. The infectious diseases consultant ceased antibiotic therapy and antifungal therapy with liposomal B amphotericin (5 mg/kg per day) was started. Due to good clinical evolution, the patient was then transferred to the cardiac surgery ward. As part of endocarditis work up, abdomen ultrasonography found no signs of splanchnic embolization while a brain CT scan revealed signs of microembolizations to the central nervous system. At this point, intravenous Isavuconazole (200 mg per day after loading doses) was added as a combination therapy with liposomal B amphotericin.

On the 10th post-operative day, a control transesophageal echocardiogram was performed showing a bioprosthetic valve with no leakage or signs of dysfunction, nor vegetations.

On the 21st post-operative day, due to confirmed good clinical evolution and negative screening blood cultures, the patient was transferred to a rehabilitation hospital. During the rehabilitation period, antifungal treatment was continued with Isavuconazole.

After 18 days in the rehabilitation hospital, the patient presented with a subtle episode of dyspnea. She was admitted to a secondary hospital and required pleural drainage due to presenting with a bilateral pleural effusion. Urgently, an 18-fluorodeoxyglucose (FDG) positron emission tomography (PET) scan was performed, revealing hypermetabolic uptake in the aortic bioprosthesis. This data was confirmed with a transesophageal echocardiogram showing the presence of a vegetation of 12 mm × 8 mm on the aortic valve bioprosthesis.

The same day, the patient was readmitted to Pisa University Hospital to be treated with surgical debridement of the prosthetic valve. Unfortunately, the patient died during the cardiac surgery re-intervention due to overwhelming vascular complications.

## 3. Materials and Methods

We conducted a Medline search for *Aspergillus* endocarditis using the terms “ASPERGILLUS” and “ENDOCARDITIS” from 1950 to October 2022; we included case reports about native-valve endocarditis and mural endocarditis in patients without previous valve surgery. All prosthetic or device-related infections were excluded. We collected 74 case reports with an available original article. Table 1 shows the baseline and microbiological features, cardiac and systemic involvement, surgical treatment and outcome (alive status) as reported by the authors. 

## 4. Discussion and Review of the Literature

This is a rare case of native-valve *Aspergillus* endocarditis in a woman without previous cardiac medical history nor ongoing immunosuppression, which is one of the main risk factors to develop invasive aspergillosis (IA) [1,2]; this is in line with our review’s finding where 64% (48 out of 74) of the patients with NVAE had underlying immunosuppressive conditions, mainly hematological malignancies (28%) and solid organ transplantation (SOT) (17%).

Nearly half of the cases of NVAE in SOT patients occurred in lung transplantation (six out of thirteen patients), which represents one of the most immunomodulated medical conditions.

We suppose that our patient’s risk factors could have been previous endothoracic surgery for neuroendocrine cancer and diabetes due to ectopic ACTH secretion. Solid tumors and diabetes are considered predisposing conditions to develop invasive fungal disease (IFD) [67,68], that in our review have been found in 4% and 5% of the patients, respectively.

With the exception of six lung-transplanted patients, in the report series, none of the patients had a history of lung surgery but 8% had undergone previous heart surgery, suggesting that thoracic intervention generally may be a risk factor to develop NVAE.

At what we know, 6% of the patients had no known comorbidity; they were all males, between 35 and 66 years of age [5,9,34,46,57].

Our patient had vegetation involving the aortic valve. We found that the aortic valve alone is less commonly involved than the mitral valve alone (13% vs. 42%, respectively) and, as a remarkable finding, native multi-valve endocarditis affected about 10% of the patients included in the review. More than half of these patients had hematological malignancies, while interestingly, about 30% of the patients had mural endocarditis, a known characteristic of AE [69,70].

Lung involvement was present in at least 40% of patients with NVAE, less than the proportion usually described in the literature about IA [1], and up to 70% of patients presented with embolization. Our patient had central nervous system (CNS) embolization and, consistently with our review, the CNS was found to be the most frequent site of embolization (54%) with a high burden of mortality with a one-year survival rate of 10%. In a teaching case, a patient that had recovered from NVAE with CNS embolization presented with a seizure 20 months after treatment, likely due to scar tissue replacement of the previous brain embolic abscess [52]. Visceral and intravascular embolization occurred in 21% and 12% of the reports, respectively.

The causative agent of our patient’s endocarditis was *A. fumigatus*, the most common cause of IA [1]. According to our review, it is the most frequent fungal isolate (58%) followed by *A. flavus* (15%) and *A. terreus* (2%). Other species found to be involved in NVAO in immunocompromised hosts were *A. niger*, *A. nidulans* and *A. udagawae*.

Diagnosis of NVAE is often difficult and 10% of the cases in this review were diagnosed at post-mortem analysis. In our case, diagnosis was made by standard microbiological culture of the valvular flap, in line with our review’s finding where in vivo diagnosis was made primarily by histological finding or cultural exam of pathological tissues. *Aspergillus* spp. were found mainly in valves (52% of cases) and emboli (16%), followed by abscesses and lung specimens such as biopsy or bronchoalveolar fluid (9%). In one case, the growth of *A. fumigatus* from pleural and pericardial fluid foretold the onset of NVAE about one year in advance prior to clinical presentation and diagnosis [43].

In our case, the blood cultures were negative; this represents an important diagnostic challenge of AE as only 10% of the patients in the review had positive blood cultures, confirming that histological examination and culture of pathological tissues are essential exams to diagnose IA [71].

In this review, since 1998, *Aspergillus* spp. galactomannan has been tested in twenty-four cases; it was positive in sixteen cases (66%), five of whom had high serum levels of 1,3-B-d-glucan. Detection of antigenemia informed clinical choices in five cases of immunocompromised hosts with no other microbiological findings and three of them survived, confirming the reliable sensitivity of these markers in the diagnosis of IA in the context of immunosuppression [32,41,55,71,72].

In at least one case, suspicion of IA was confirmed by molecular methods on blood samples but the role of this method in the diagnosis of IA is still uncertain due to the lack of standardization [66,71]; an interesting use of polymerase chain reaction on blood samples is suggested in one case where it has been used to monitor antimycotic drug efficacy in a pediatric liver transplant recipient who had a favorable outcome without cardiac surgery [32].

In this review, NVAE has been confirmed to be a life-threatening condition with an overall mortality rate of at least 60%.

Our patient underwent surgical valve replacement. In our review, surgical treatment was performed in 58% of patients with a higher survival rate in patients that underwent cardiac surgery rather than those who did not receive surgical treatment (34% vs. 22%, respectively). It was performed for 15 out of the 22 (68%) surviving patients. These findings are in line with most important guidelines that recommend surgical intervention in cases of fungal endocarditis [73,74].

Triazoles and polyenes are the main medical options for IA and their use is strictly tied to fungal species, drug susceptibility, host characteristics and therapeutic drug monitoring [71].

On the basis of fungal growth on the valvular flap, an antimycotic therapy with liposomal B amphotericin was initially started and then implemented adding isavuconazole after the radiological finding of microembolization to the CNS. In our review, these antifungal agents were used in the same patient in eleven out of the twenty-two cases that had favorable outcomes, three of whom had not undergone surgical intervention; in eleven cases, voriconazole was used alone and in six of these cases, patients survived at 1 year, including a pediatric patient who did not undergo cardiac surgery and another patient reported to be on undefined oral therapy, while monotherapy of amphotericin B was administered in fifteen reports, nearly half of whom in the pre-liposomal and pre-triazoles era, with only two patients surviving [36,63].

In our review, eight cases of diagnosed IFD that underwent therapy occurred before 1997, a year during the early triazole era [75] in which the Food and Drug Administration (FDA) approved the commercialization of liposomal B amphotericin formulations [76], and only one patient survived. Since then, the overall survival rate in our review increased up to 35%.

Our patient underwent surgical valve replacement while not on antifungal therapy and died due to a relapse on the bioprosthesis while she was still taking appropriate antifungal therapy. Valve surgery is one of the main predisposing conditions to develop AE and in our review, at least four patients had relapses on their bioprostheses with unfavorable outcomes, three of whom were not on antimycotic therapy at the moment of surgical intervention. This finding, associated to our report, hints that prompt antifungal therapy may reduce the risk of relapses in AE, a high mortality event that can occur even more than one year from valve surgery [3,30].

## 5. Conclusions

*Aspergillus* endocarditis can occur on native valves and it is a life-threatening condition that requires prompt diagnosis. It should be suspected in negative blood culture endocarditis both in patients with underlying immunosuppressive conditions and in immunocompetent hosts. Diagnosis is not easy; the clinical context and biomarkers can guide the suspicion but confirmation with tissue histology and culture is fundamental.

The optimal approach is early treatment with surgical debridement and appropriate antifungal agents, mainly triazoles. Antifungal prophylaxis may be a strategy to reduce the incidence of this condition in immunocompromised patients with negative blood culture endocarditis that undergo valve surgery. Follow-up imaging after cardiac surgery should include a cardiac PET-CT scan at 3–6 months after the procedure, in order to decide safely to stop or continue the antifungal maintenance regimen. In some cases, life-long maintenance antifungal treatment is necessary. Despite the improvement of both diagnostic methods and antifungal drugs, timely diagnosis and appropriate treatment remains a major challenge of this fatal disease.

## Figures and Tables

**Table 1 antibiotics-12-01190-t001:** Case reports description.

Ref.	Year	Author	Age	Gender	Comorbidities	Cardiac Involvement	Aspergillus Species	Microbiological Positive Tissue	Lung Involvement	Embolization	Surgery	Alive
[4]	1979	Mikulski	49	M	ALL	Mural, aortic, mitral	*A. flavus*	Valve	Yes	Kidney, spleen	No	No
[5]	1983	Vishnniavsky	66	M	No	Mitral	*A. fumigatus*	Valve	No	Eye, femoral artery	Yes	No
[6]	1986	Nishiura	68	M	ALL	Mitral	-	Post-mortem histological examination: heart and lungs	Yes	No	No	No
[7]	1989	Woods	19	M	SOT (liver)	Tricuspid	*A. flavus*	Valve	Yes	CNS, lungs, mesentery	No	No
[8]	1990	Cox	32	M	AIDS	Mural, mitral	*A. fumigatus*	Post-mortem histological examination: heart	No	CNS, kidney, spleen, pancreas	No	No
[9]	1990	Bogner	74	M	No	Aortic	*A. fumigatus*	Blood	No	Hand	No	Yes
[10]	1991	Light	55	M	IVDU	Mitral	*A. flavus*	Post-mortem histological examination: heart and embolus	No	CNS, kidney, spleen, iliac and femoral arteries	No	No
[11]	1996	Casson	3	F	CGD	Mural	*A. nidulans*	Valve	No	Skin	Yes	No
[12]	1996	Sergi	55	M	ABSSSI	Mural	-	Post-mortem histological examination: heart	Yes	Thyroid, kidney	No	No
[13]	1997	Kanda	46	M	CML	Mitral	-	Post-mortem histological examination: heart and CNS	Yes	CNS, aorta, spleen	No	No
[14]	1997	Khan	64	M	Amebiasis	Mitral	*A. flavus*	Valve	No	Kidney	Yes	No
[15]	1998	Kennedy	6	M	Solid tumor (neuroblastoma)	Mural, tricuspid	*A. flavus*	Blood	No	No	No	Yes
[16]	1998	Schett	60	F	ALL	Mural, mitral	*A. terreus*	Valve	No	Spleen, kidney	Yes	No
[17]	1999	Marin	32	M	SOT (kidney)	Mitral	*A. fumigatus*	Embolus	No	Kidney, lower limb	No	No
[18]	2000	Gilbey	31	M	CF, SOT (lungs)	Mitral	*A. fumigatus*	Embolus	Yes	Brachial artery, skin	No	No
[19]	2000	Gumbo	67	M	Bronchiectasis	Mitral	*A. fumigatus*	Valve, abscess	No	CNS	Yes	No
[19]	2000	Gumbo	57	F	BOOP, bronchiectasis	Mitral	*A. fumigatus*	Valve, abscess	No	CNS, kidney, spleen, skin, thyroid, femoral artery	Yes	No
[20]	2000	Rao	11	M	AML	Mural, mitral, tricuspid	*A. flavus*	Valve	No	Skin	No	Yes
[19]	2000	Gumbo	53	M	CLL	Aortic	*A. fumigatus*	Valve	No	Spleen, ocular	Yes	No
[21]	2001	Petrosillo	39	M	IVDU, HIV	Aortic	-	Post-mortem histological examination: heart	Yes	No	No	No
[21]	2001	Petrosillo	35	M	IVDU, HIV	Tricuspid	*A. fumigatus*	Post-mortem histological examination: valve	No	No	No	No
[21]	2001	Petrosillo	31	M	IVDU	Aortic	*A. niger*	Embolus	No	CNS, iliac artery	Yes	No
[22]	2004	Kotanidou	61	F	TTP	Mitral	*A. fumigatus*	Valve	No	CNS, spleen, kidney	Yes	No
[23]	2004	Shoar	19	F	Previous heart surgery	Tricuspid	*A. flavus*	Valve	Yes	CNS	No	No
[24]	2004	Verghese	34	M	Previous heart surgery	Aortic	*A. terreus*	Embolus	No	Femoral artery	Yes	No
[25]	2005	Scherer	28	F	CF, SOT (lungs)	Mitral	*A. fumigatus*	Valve	No	CNS, ocular	Yes	No
[26]	2006	Petrikkos	17	M	Aplastic anemia	Mitral	-	Post-mortem histological examination: heart, lung and CNS	Yes	CNS	No	No
[26]	2006	Petrikkos	31	F	Aplastic anemia	Mural, aortic, mitral	*A. fumigatus*	Lung	Yes	CNS, spleen, kidney, skin	No	No
[27]	2006	Vassiloyanakopoulos	25	M	Asthma	Tricuspid	*A. fumigatus*	Valve	Yes	No	Yes	Yes
[28]	2007	Peman	58	M	COPD	Mitral	*A. fumigatus*	Valve and blood	No	CNS, kidney, mesenteric artery	Yes	No
[29]	2007	Saxena	57	M	SOT (lungs)	Mitral	*A. fumigatus*	Valve	Yes	No	Yes	Yes
[30]	2008	Maher	19	F	CF, SOT (lungs)	Aortic	*A. fumigatus*	Valve and abscess	Not clear	Aorta	Yes	No
[30]	2008	Maher	24	F	CF, SOT (lungs)	Mitral	*A. fumigatus*	Valve	No	Mesentery	Yes	No
[31]	2008	Morio	53	M	SOT (heart)	Mitral	*A. fumigatus*	Valve	Yes	CNS	Yes	No
[32]	2008	Mourier	8	F	SOT (liver)	Mitral	*A. fumigatus*	-	Yes	-	No	Yes
[33]	2009	Esmaelizadeh	49	M	CABG	Aortic	-	Valve	No	Aorta	Yes	No
[34]	2009	Ryu	35	M	None	Mitral	-	Valve	No	Femoral artery, mesenteric artery	Yes	Yes
[35]	2010	Gupta	60	M	Asthma, ABPA	Tricuspid, pulmonary, mural	*A. fumigatus*	Post-mortem histological examination: heart	Yes	Kidney, colon, thyroid	No	No
[36]	2010	Kalokhe	18	M	HSCT	Tricuspid, device	*A. fumigatus*	Valve	No	No	Yes	Yes
[37]	2011	Lazaro	40	M	SOT (lung)	Mural, mitral	*A. fumigatus*	Valve	Yes	CNS, kidney, skin, limbs	Yes	No
[38]	2011	Nikoulosis	11	M	ALL	Mural	*A. fumigatus*	Valve	Yes	No	Yes	Yes
[38]	2011	Nikoulosis	2	F	ALL	Mural	*A. fumigatus*	Blood	No	No	No	Yes
[39]	2011	Palomares	66	M	CLL	Mitral	*A. fumigatus*	Valve	No	No	No	No
[40]	2012	Attia	60	F	CLL	Mural, aortic	*A. fumigatus*	Valve	Yes	No	Yes	No
[41]	2012	Kuroki	68	M	Solid tumor (colon)	Mitral	*A. fumigatus*	-	Yes	CNS	Yes	Yes
[42]	2013	Ansari	51	M	AML	Tricuspid	-	Valve and embolus	Yes	Ocular and thrombophlebitis	Yes	Yes
[43]	2013	Regueiro	21	F	CF, SOT (lungs)	Aortic	*A. fumigatus*	Valve, embolus and pericardial liquid	No	Lower limb	Yes	Yes
[44]	2013	Vohra	53	M	CABG, diabetes	Tricuspid	*A. fumigatus*	Lung	Yes	No	No	No
[45]	2014	Seo	53	M	AML	Mitral	-	Post-mortem histological examination: embolus	No	Coronary	Yes	No
[46]	2015	Demirsoy	36	M	HSCT	Mitral	*A. flavus*	Valve	Yes	CNS, skin	Yes	No
[47]	2015	Olszanecka	33	M	Bone marrow aplasia	Mural, aortic, mitral	-	-	.	Aorta	No	No
[48]	2016	Ikediobi	65	M	None	Aortic, mitral	*A. fumigatus*	Valve	No	No	Yes	No
[49]	2016	Rofaiel	64	F	APL	Mitral	*A. fumigatus*	Valve	No	CNS	Yes	No
[50]	2017	Seki	70	M	HSCT, diabetes	Mitral	*A. udagawae*	Valve	No	CNS, kidney, spleen, stomach, bladder	Yes	No
[51]	2018	Alsobayeg	56	F	SOT (kidney-liver), diabetes	Aortic, mitral, tricuspid	*A. flavus*	Valve	No	No	Yes	No
[52]	2018	Marinelli	39	M	Alcohol consumption, nocardiosis	Mural, tricuspid	*A. fumigatus*	Abscess	No	CNS	No	Yes
[53]	2018	Sabir	30	M	Rheumatic heart disease	Mitral	-	Valve	No	No	Yes	Yes
[54]	2018	Varghese	65	M	HLH	Mural, mitral	-	-	Yes	CNS	No	No
[55]	2018	Yamamoto	38	F	SLE	Mural	-	-	No	CNS	Yes	Yes
[56]	2019	Clarke	63	F	CABG	Aortic	*A. flavus*	Embolus	No	Femoral artery	Yes	Yes
[57]	2019	Hatlen	58	M	None	Mitral	*A. fumigatus*	Blood	No	CNS	Yes	No
[58]	2020	Aggarwal	49	M	Trauma, splenectomy	Mitral	*A. fumigatus*	Valve	Yes	CNS	Yes	No
[59]	2020	Aldosari	71	M	Solid tumor (prostate), diabetes, rheumatic disease	Mitral	*A. fumigatus*	Valve and abscess	No	CNS	Yes	Yes
[60]	2020	Fattahi	4	F	Marfan syndrome	Mural, mitral	*A. flavus*	Valve	No	No	Yes	Yes
[61]	2020	Nanditha	49	M	Obesity, previous pulmonary embolism	Aortic	*A. fumigatus*	Abscess	No	No	Yes	Yes
[62]	2021	Al Mashdali	49	M	CABG, diabetes	Aortic	*A. fumigatus*	Valve	No	Spleen, CNS	Yes	No
[63]	2021	Anjani	6	M	CGD	Mitral	*A. fumigatus*	Abscess	No	Skull bone	No	Yes
[64]	2021	Carvalho	47	F	Seronegative spondyloarthropathy	Mitral	-	Bronchoalveolar lavage	Yes	CNS	No	No
[65]	2021	Chevalier	71	F	Chronic neutropenia	Mitral	*A. fumigatus*	Bronchoalveolar lavage	Yes	CNS	No	No
[65]	2021	Chevalier	52	F	AIDS	Mural, mitral	*A. fumigatus*	Bronchoalveolar lavage	Yes	No	No	No
[65]	2021	Chevalier	58	M	CLL	Mural	*A. fumigatus*	Post-mortem histological examination: heart	Yes	No	No	No
[65]	2021	Chevalier	55	M	IgA vasculitis	Mural, mitral	*A. fumigatus*	Embolus	Yes	CNS, ocular	Yes	No
[65]	2021	Chevalier	58	M	SOT (kidney)	Mural, aortic	*A. fumigatus*	Bronchoalveolar lavage and abscess	Yes	CNS, muscle	No	No
[66]	2021	Dai	36	F	Hemodialysis	Tricuspid	*A. fumigatus*	Valve	Yes	No	Yes	No

Legenda: ALL: acute lymphoblastic leukemia; CNS: central nervous system; SOT: solid organ transplantation; AIDS: acquired immune deficiency syndrome; IVDU: intravenous drug user; CGD: chronic granulomatous disease; ABSSSI: acute bacterial skin and skin structure infection; CML: chronic myeloid leukemia; CF: cystic fibrosis; BOOP: bronchiolitis obliterans organizing pneumonia; AML: acute myeloid leukemia; CLL: chronic lymphocytic leukemia; HIV: human immunodeficiency virus; TTP: thrombotic thrombocytopenic purpura; COPD: chronic obstructive pulmonary disease; CABG: coronary artery bypass graft surgery; ABPA: allergic bronchopulmonary aspergillosis; HSCT: hematopoietic stem cell transplantation; APL: acute promyelocytic leukemia; HLH: hemophagocytic lymphohistiocytosis; SLE: systemic lupus erythematosus.

## Data Availability

Not applicable.

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
