# Peer review of "Native-Valve Aspergillus Endocarditis: Case Report and Literature Review"

_antibiotics, 2023, doi:10.3390/antibiotics12071190_

Round 1

Reviewer 1 Report

Regarding the diagnosis of infective endocarditis on the native valve, I would like to know if the patient had echocardiography before this episode of the disease since a degenerative lesion of the aortic valve cannot be excluded due to the risk factors present (smoking, dyslipidemia, diabetes, hypertension)

According to the ESC guideline, the empiric treatment for IE with negative blood cultures is vancomycin with gentamicin and rifampicin. In the presented clinical case, treatment was initiated with Meropenem and Vancomycin. What was the reason for this decision?

For Aspergillus EI, voriconazole is the drug of choice, and some experts recommend the combination of amphotericin B or echinocandin. In the presented case, treatment with amphotericin B was started. What was the reason for this decision?

I would like the authors to state in the article whether there are accurate data proving that microembolization in the central nervous system is not thrombotic in nature and are infectious in nature. I would also suggest that the article mention what type of anticoagulant treatment the patient was given, for what period of time, and whether the patient was in sinus rhythm during this period of time.

The detection of galactomannan antigen has proven effective both in establishing the diagnosis and in assessing the prognosis in the case of Aspergillus infection. I would like to know if the galactomannan antigen test was performed during the evolution of the case, when it was decided to continue the antifungal medication only with Isavuconazole after this drug was associated for two weeks with amphotericin B. Isavuconazole was not inferior to voriconazole for the treatment of Aspergillus infection.

I commend the authors for the meticulous and complex work they have done to research past and present Aspergillus endocarditis.

Author Response

1)Regarding the diagnosis of infective endocarditis on the native valve, I would like to know if the patient had echocardiography before this episode of the disease since a degenerative lesion of the aortic valve cannot be excluded due to the risk factors present (smoking, dyslipidemia, diabetes, hypertension) A: no, the patient had not echocardiography before the acute episode.

According to the ESC guideline, the empiric treatment for IE with negative blood cultures is vancomycin with gentamicin and rifampicin. In the presented clinical case, treatment was initiated with Meropenem and Vancomycin. What was the reason for this decision?A: this treatment was started by the Intensivologist and maintened by the cardiac surgeons before the ID team could make a consultation

For Aspergillus EI, voriconazole is the drug of choice, and some experts recommend the combination of amphotericin B or echinocandin. In the presented case, treatment with amphotericin B was started. What was the reason for this decision? A: Amphotericin B was preferred due to the superiority in terms of tolerability and above all lacking TDM for Voriconazole in our hospital

I would like the authors to state in the article whether there are accurate data proving that microembolization in the central nervous system is not thrombotic in nature and are infectious in nature. I would also suggest that the article mention what type of anticoagulant treatment the patient was given, for what period of time, and whether the patient was in sinus rhythm during this period of time. A: the septic nature of the microembolization has been reported by the neuroradiologist reviewing the features of the brain CT with contrast. The patient received standard post-operative LMWH. 

The detection of galactomannan antigen has proven effective both in establishing the diagnosis and in assessing the prognosis in the case of Aspergillus infection. I would like to know if the galactomannan antigen test was performed during the evolution of the case, when it was decided to continue the antifungal medication only with Isavuconazole after this drug was associated for two weeks with amphotericin B. Isavuconazole was not inferior to voriconazole for the treatment of Aspergillus infection.

A: galactromannan unfortunately, due to lack of reagents, was not available in that period in our hospital. For sure It would helped us a lot. Thanks for the question!

I commend the authors for the meticulous and complex work they have done to research past and present Aspergillus endocarditis. 

Reviewer 2 Report

The paper is an interesting and informative reminder of a rare form of infective endocarditis caused by fungi.

.

Author Response

Thanks, we did the best we could with available. data

Reviewer 3 Report

In this narrative review, the authors discuss infective endocarditis caused by Aspergillus species. Despite covering the period from 1950 to October 2022 and featuring 74 case reports, Table 1 only includes 33 case reports that conclude in 2008. On the other hand, References numbers 31 to 66, featuring cases reported between 2009 and 2021, are not reflected in Table 1. Consequently, it is crucial to examine Table 1 thoroughly.

Author Response

Thanks for finding this problem. The original table was cut. 

Round 2

Reviewer 3 Report

No comments or suggestions at this point.